# Identification and Structural Aspects of G-Quadruplex-Forming Sequences from the Influenza A Virus Genome

**DOI:** 10.3390/ijms22116031

**Published:** 2021-06-02

**Authors:** Maria Tomaszewska, Marta Szabat, Karolina Zielińska, Ryszard Kierzek

**Affiliations:** 1Department of Structural Chemistry and Biology of Nucleic Acids, Institute of Bioorganic Chemistry, Polish Academy of Sciences, Noskowskiego 12/14, 61-704 Poznan, Poland; mtomaszewska@ibch.poznan.pl; 2Department of Biomolecular NMR, Institute of Bioorganic Chemistry, Polish Academy of Sciences, Noskowskiego 12/14, 61-704 Poznan, Poland; kczajczynska@ibch.poznan.pl

**Keywords:** Influenza A virus, potential quadruplex-forming sequence, RNA G-quadruplexes

## Abstract

Influenza A virus (IAV) causes seasonal epidemics and sporadic pandemics, therefore is an important research subject for scientists around the world. Despite the high variability of its genome, the structure of viral RNA (vRNA) possesses features that remain constant between strains and are biologically important for virus replication. Therefore, conserved structural motifs of vRNA can represent a novel therapeutic target. Here, we focused on the presence of G-rich sequences within the influenza A/California/07/2009(H1N1) genome and their ability to form RNA G-quadruplex structures (G4s). We identified 12 potential quadruplex-forming sequences (PQS) and determined their conservation among the IAV strains using bioinformatics tools. Then we examined the propensity of PQS to fold into G4s by various biophysical methods. Our results revealed that six PQS oligomers could form RNA G-quadruplexes. However, three of them were confirmed to adopt G4 structures by all utilized methods. Moreover, we showed that these PQS motifs are present within segments encoding polymerase complex proteins indicating their possible role in the virus biology.

## 1. Introduction

Influenza virus, due to its pandemic potential, became an interesting subject in various biological research projects. There are three types of influenza virus: A, B and C. However, the influenza A virus (IAV) is gaining a lot of attention because of its responsibility for most of the pandemic outbreaks [1], the ability to infect fowl, humans and other mammals, and the highest zoonotic potential allowing the virus to cross the barrier from animals to humans [2]. This is possible due to the reassortment (antigenic shift) of the viral RNA (vRNA) segments during the co-infection with different IAV strains [3]. Besides this ability, there are other IAV traits that make it such a serious threat to public health, i.e., easy transmission, fast adaptation to the particular host immune system, and unceasing evolutionary dynamics correlated with appearing point mutations in the viral genome (antigenic drift). Accumulation of these mutations can eventually lead to the emergence of the virus with novel antigenic properties that can evade the host immune response [4]. Therefore, it is important to discover novel therapeutic strategies and anti-influenza drugs with a new mode of action.

The IAV negative-sense RNA genome consists of eight segments encoding 11 proteins: three polymerase complex proteins: polymerase basic protein 1 (PB1), polymerase basic protein 2 (PB2) and polymerase acidic protein (PA), hemagglutinin (HA), nucleoprotein (NP), neuraminidase (NA), two matrix proteins (M1 and M2) and non-structural proteins (NS1, NS2 and PB1-F2) [5]. Interestingly, each of the vRNA segments with nucleoprotein and polymerase subunits forms a viral ribonucleoprotein (vRNP) complex being a basic element during genome replication and transcription [6]. Overall, vRNP structure is related to its functions. It was reported that the viral replication, RNA packaging, mRNA splicing regulation or recognition by the host immune system are controlled by the RNA structure [7,8]. Within the vRNP complex, NP molecules bind vRNA with high affinity and serve as the regulator of the nuclear export and import of vRNP. However, it was previously confirmed that vRNA can escape from complex with NP and potentially fold into the secondary structure in a dynamic manner [9]. Recently, there has been increasing research indicating the crucial role of viral genome secondary structure in the replication and other biological processes [7], not only in the case of the influenza virus [10,11], but also regarding other viruses. For instance, human immunodeficiency virus (HIV) vRNA structure can regulate genomic RNA transcription and gene expression that affect the cellular innate immunity [12].

Importantly, there is growing interest in the unique non-canonical structures called G-quadruplexes (G4s) [13,14]. They can be formed within guanine-rich (G-rich) sequences called potential quadruplex-forming sequences (PQS). The G-quadruplex involves G-tetrads composed of a planar array of four guanine residues associated through Hoogsteen hydrogen bonds. The G-tracts are connected with each other by different kinds of loop indicating the G-quadruplex folding topology. Depending on the sequence, length of the loops and strand orientation, the G4 structures can be parallel, antiparallel or hybrid type (3 + 1) [15]. Moreover, the number of molecules involved in the G4s formation can influence the folding topology. G-quadruplex structures can be intramolecular meaning that one strand folds within itself to form the G-tetrads or intermolecular consisting of two or more strands [16]. Previously published studies revealed the presence of G4 structures in the genomes of RNA viruses, e.g., Zaire ebolavirus (EBOV) [17], Zika virus [18] and more recently also in severe acute respiratory syndrome coronavirus 2 (SARS-CoV-2) [19] as well as in the IAV [20]. Apparently, these structures seem to be highly conserved in the viral genomes, indicating their important role in the viral life cycle, for instance during replication. These findings encouraged us to investigate the IAV genome for the presence of the unique G-rich sequences.

In the current study, both bioinformatics analysis and biophysical methods allowed us to show that conserved PQS motifs occur within the influenza A/California/07/2009(H1N1) genome and indeed have the ability to fold into G-quadruplexes. Based on these results and taking under consideration the G4s localization within the vRNA segments, it can be assumed that these structures can play an important role in the regulation of different steps of the viral life cycle.

## 2. Results

### 2.1. Identification of Potential Quadruplex-Forming Sequences (PQS) Motifs in the Influenza A Virus (IAV) Genome

In this work, we inspected the IAV genome for the presence of the unique G-rich sequences. For this purpose, the sequence of the influenza A/California/07/2009(H1N1) genome was retrieved from the NCBI (National Center for Biotechnology Information) database (RefSeq assembly accession: GCF_001343785.1). There are several bioinformatics tools allowing to search for G-rich regions in RNA sequences prone to fold into G-quadruplex structures [21]. Herein, G4RNA screener (accessed on 15 December 2020) (http://scottgroup.med.usherbrooke.ca/G4RNA_screener) and the Quadruplex-forming G-Rich Sequences (QGRS) Mapper (accessed on 15 December 2020) (http://bioinformatics.ramapo.edu/QGRS/analyze.php) were used to identify PQS motifs across the whole IAV genome. Our bioinformatics analysis revealed the presence of twelve PQS candidates that can potentially adopt RNA G4 structures. The PQS motif characteristics are summarized in Table 1. Interestingly, all PQS candidates contain three or more of at least two continuous guanosine residues (Gs) within a sequence and a half of them contains four (1KW, 3KW, 6KW, 7KW and 10KW) or five (12KW) continuous Gs (Table 1). Therefore, it is plausible that these sequences have the potential to form RNA G-quadruplex structures.

### 2.2. Identification of Nucleotide Residues Frequency in PQS Motifs

All PQS motifs found were located within the protein-coding regions of the IAV genome. Obtained results encouraged us to ask the question of whether these PQS candidates are conserved among the IAV strains. To answer this question, we first used the IRD (Influenza Research Database) database (accessed on 15 December 2020) (www.fludb.org) to retrieve complete genomic sequences of the eight IAV segments. Subsequently, the multiple sequence alignments were carried out for each segment separately, and the frequency of nucleotide residues at each position was established using MAFFT software and BioEdit 7.1 program, respectively. The percentage of the PQS motifs similarity across the IAV strains was calculated and is presented in Table 2.

This comparative analysis showed that similarity of PQS candidates ranged from 71.0% to 97.5% and from 86.0% to 99.6% for the influenza type A and the H1N1 subtype, respectively (Table 2). As expected, the higher percentage of similarity was obtained for the strains of the H1N1 subtype compared to all the IAV strains. Interestingly, the only exception was observed in the case of 7KW variant, for which the result was opposite (similarity was higher within the influenza type A than within the H1N1 subtype, 96.1 vs. 93.4%, respectively).

Finally, the conservation level of PQS motifs within the H1N1 subtype was estimated using WebLogo application [22,23]. More specific, all PQS motifs were searched in aligned sequences and LOGO representation was generated (Table 3). The number of aligned sequences used for this analysis varied for each vRNA segments and was presented above in Table 2. Based on our results we observed that the conservation levels were different depending on PQS motifs and ranged from 44% for 4KW to 92% for 12KW (Table 3). However, most of PQS motifs showed the conservation levels between 75% and 83.4% (1KW-3KW, 5KW, 7KW–9KW, Table 3). This indicates that some of the sequences are highly conserved, e.g., 12KW (92%, Table 3), whereas 4KW (44%, Table 3) is significantly less conserved within the analyzed sequences of H1N1 subtype. Interestingly, when we take under consideration the conservation pattern of nucleotides, we can conclude that G residues show relatively high conservation within the G-tracts of PQS motifs. The average conservation level in this position (G-rich regions) was 69.8%. Obtained data suggest that these regions can be important in genome organization of the IAV.

Noticeably, the formation of G4 structures by PQS candidates cannot be unequivocally assumed without experimental validation. Therefore, we chemically synthesized RNA oligomers and used different types of spectroscopic techniques (including nuclear magnetic resonance (NMR), ultraviolet–visible (UV-Vis) melting, circular dichroism (CD) and fluorescence assays) as well as native gel electrophoresis to examine the structure of PQS motifs identified in our bioinformatics analysis

### 2.3. Characterization of PQS Motifs by Nuclear Magnetic Resonance (^1^H NMR)

NMR spectroscopy is a powerful tool to study nucleic acids secondary structures including G-quadruplex formation [24,25]. By recording ^1^H NMR spectra of DNA or RNA samples, we can gain information whether the molecule forms quadruplex structure under appropriate solution conditions. The core of the quadruplexes are G-tetrads in which guanosine residues are associated through Hoogsteen hydrogen bonds. Signals from the imino protons involved in this type of bond are usually in the area of 10–12 ppm, whereas those from 12–15 ppm are characteristic for Watson–Crick base pairs. The imino proton region of NMR spectra between 10 and 15 ppm provides information about the nature of hydrogen bonds between DNA or RNA base pairs [15].

Hence, we first performed ^1^H NMR experiments to justify the selection of RNA G4-forming motifs. Our NMR analysis of PQS candidates provided direct evidence for the formation of RNA G4 structures by several oligomers. Based on the results obtained, the model PQS oligomers were divided into three groups depending on their predicted structure. The first group includes sequences with the highest propensity to fold into G-quadruplex structures, including 1KW, 7KW and 11KW variants (Figure 1a). Obtained ^1^H NMR spectra for these PQS displayed the imino proton peaks between 10 and 12 ppm, which are characteristic for Hoogsteen base pairs. The second group contains three PQS oligomers (3KW, 6KW and 9KW) that might be in dynamic equilibrium between two different structures, including RNA G-quadruplex and hairpin. Their ^1^H NMR spectra demonstrated the signals within the range of 10–12 ppm, suggesting Hoogsteen hydrogen bonds involved in G4 structure formation (Figure 1b). Additionally, the presence of imino resonances found between 12 and 14 ppm supports the Watson–Crick base pairing and possible hairpin structure formation (Figure 1b). These results can indicate that 3KW, 6KW and 9KW variants have lower propensity to form RNA G4 structures in comparison with PQS oligomers assigned to the first group.

On the other hand, the third group of PQS oligomers does not form stable RNA G4 structures. It can be indicated that 2KW, 4KW, 5KW, 8KW, 10KW and 12KW variants adopt a hairpin structure. The structure of these PQS oligomers was predicted using RNAstructure and is shown in Appendix A.

Additionally, we performed NMR experiments with the addition of N-methyl mesoporphyrin IX (NMM) that is a dye selectively binding to G-quadruplexes. Experiments were conducted for PQS oligomers with the addition of NMM in a 1:1 molar ratio. In general, the recorded ^1^H NMR spectra for the first and the second group of PQS displayed broad and shifted signals between 10–12 ppm, indicating binding of NMM to the G4 structure. It is an additional confirmation of RNA G-quadruplex formation by few PQS candidates.

### 2.4. Determination of Thermal Melting Profiles of PQS Motifs

UV spectroscopy is a fast and simple technique to confirm G4 structure formation and evaluation of its thermal stability. Typically, the G-quadruplex unfolding is associated with a decrease in absorbance at 295 nm wavelength, at which the thermal denaturation of the G4 structure is manifested in an over 50% change in the absorbance amplitude [26,27]. Recording the profile of the melting curve at this wavelength results in hypochromic effect (decrease in absorbance with increasing temperature) [15,26,27]. Therefore, despite the fact that we performed measurements at two wavelengths: 260 and 295 nm, we decided to study the RNA G-quadruplex formation by analysis of the melting profile at 295 nm.

As a result, we were not able to obtain reliable thermodynamic parameters and decided to focus on the melting profiles and changes in the melting temperature determined for model PQS variants. The obtained results demonstrated that the melting curve profiles of four PQS oligomers (1KW, 3KW, 7KW and 11KW, Figure 2) were inverted, indicating RNA G4 formation. Unfortunately, we did not observe the hypochromic effect at 295 nm for remaining PQS variants, which suggests that these structures have lower thermal stability or an equilibrium between different folding forms exists under experimental conditions. Based on the thermal denaturation measurements we calculated the melting temperature values (T_m_) that are presented in Figure 2.

Based on the data obtained we were able to determine the T_m_ values for only four PQS oligomers (Figure 2). Individually, the melting analysis revealed that the most stable variant was 1KW (T_m_ = 50.7 °C, Figure 2), whereas variant 11KW was characterized by the lowest value of T_m_ parameter (28.6 °C, Figure 2). The 3KW and 7KW variants were less stable in comparison to 1KW and their melting temperatures were 48.6 °C and 44.6 °C, respectively (Figure 2). These results can indicate that the changes in thermal stability of studied PQS oligomers are correlated with the number of G-tracts in a strand. The 1KW oligomer contains two tracts of four continuous Gs and one tract of two continuous Gs, whereas the least stable 11KW variant has one tract of three continuous Gs and three tracts with two continuous Gs.

### 2.5. Circular Dichroism Spectra of PQS Oligomers

Circular dichroism (CD) spectroscopy is a simple and relatively fast biophysical method to characterize the G-quadruplex folding topology and assess its structural geometry. It is based on a phenomenon that chiral molecules absorb left-handed and right-handed circularly polarized light differently [28]. Generally, CD spectra of nucleic acids display characteristic peaks in the spectral range from 200 nm to 320 nm and are sensitive to base-stacking interactions. Due to the fact that G4 structures can adopt three topologies characterized by differences in tetrad stacking, we can observe a unique CD signature for a given topology [15,29]. Therefore, CD spectroscopy was employed herein to evaluate the ability of PQS motifs to form RNA G4s. The resultant CD spectra for these PQS oligomers displayed the profile typical for a parallel G-quadruplex structure (Figure 3).

One negative band around 240 nm and one positive band near 265 nm were observed for studied PQS variants (Figure 3). Surprisingly, we observed an additional negative peak around 295 nm in case of 6KW variant that is difficult to interpret at this moment. Nevertheless, results obtained stay in accordance with previously published data regarding the formation of parallel-stranded G-quadruplexes from the viral genomes [17,18,19]. Interestingly, most of the RNA G4s adopt a parallel strand orientation, however, the first evidence of an antiparallel RNA G-quadruplex formed by human telomere RNA was recently described by Bao and Xu [30].

### 2.6. Thioflavin T Fluorescence Intensity Measurements

To further confirm that PQS motifs adopt a G-quadruplex structure, thioflavin T (ThT) fluorescence intensity measurements were performed. In 2013 ThT was applied as a specific ligand towards G-quadruplex folds [31]. Because the fluorescence enhancement results from the specific binding of ThT to G4 structure, the ThT probe was utilized for selective detection of RNA G-quadruplexes by many research groups [18,32,33]. Hence, in our studies we expected to provide additional RNA G-quadruplex confirmation by conducting ThT fluorescence assays. More specifically, we used two oligomers (2KW and 4KW from the IAV genome that failed to adopt a RNA G-quadruplex in previous experiments) as negative controls and a sequence from the Ebola virus genome examined earlier by Wang et al. [17] as a positive control, named 17KW 5′r(GGGGUCAUAUGGGAGGGAUUGAAGG). In addition, the buffer and ThT alone were used as negative controls. ThT was directly added to oligonucleotide solutions and the fluorescence measurements were performed on a microplate reader.

As a result, the fluorescence emission spectra of ThT in the presence of various RNA oligomers at wavelengths ranging from 448 nm to 700 nm and the fluorescence emission of ThT at 482 nm are presented in Figure 4a,b, respectively. Based on the results, the maximum fluorescence peak was recorded around 490 nm in all cases (Figure 4a). In addition, large increments in the fluorescence intensity for all of the RNA oligomers from the first group (1KW, 7KW and 11KW), two oligomers from the second group (3KW and 6KW) and positive control (17KW) compared with negative control probes (2KW and 4KW) were observed (Figure 4). Interestingly, we observed the fluorescence enhancement of ThT also in the presence of the 5KW variant which was assigned to third group (non-G-quadruplex forming sequence) based on the NMR analysis. This can suggest that 5KW oligomer may adopt the G4 structure under certain conditions. As expected, 9KW variant was the only exception showing no significant increase in ThT fluorescence indicating that this oligomer does not form a G-quadruplex structure, which is consistent with the UV melting analysis.

Comparing the fluorescence intensity at 482 nm for tested PQS variants we found a significant increase in ThT fluorescence upon binding to almost all of them (1KW, 7KW, 11KW, 3KW, 6KW and 5KW, Figure 4b). Obtained data showed that these oligomers yielded substantially higher fluorescence enhancement (except 9KW variant) in reference to the 2KW and 4KW selected as non-G-quadruplex forming oligomers. According to these results, we observed that the fluorescence enhancement at 482 nm was 728-, 564- and 537-fold for 11KW, 17KW and 6KW, respectively, compared with the signal of ThT alone (Figure 4b) supporting RNA G-quadruplex formation. Moreover, the substantial increase in the ThT fluorescence emission compared to ThT alone was recorded in the presence of three RNA oligomers (3KW, 5KW and 7KW) and was 466-, 435- and 323-fold, respectively (Figure 4b). These findings indicate that studied PQS variants adopt RNA G-quadruplex structures. In contrast, the 9KW variant did not induce a significant fluorescence increase (only a 78-fold enhancement, Figure 4b), which suggests adopting structure other than G4s. It is worth noting that ThT exhibited higher fluorescence enhancement in the presence of 11KW variant than for our positive control, 17KW (728-fold vs. 564-fold, Figure 4b). This indicates that ThT ligand has strong binding affinity to 11KW variant and supports assumption that this PQS folds into the RNA G4 structure. Taking under consideration the above data and the previously described results [17,18,31,34], it can be concluded that the ThT fluorescence signal may be used to predict G-quadruplex formation and depends on several factors, e.g., nucleotide sequence or available binding sites.

### 2.7. Native Gel Electrophoresis

Finally, native polyacrylamide gel electrophoresis (PAGE) was performed to confirm the RNA G4s formation. DNA or RNA migrates at different rates in an electric field depending on its secondary structure. Moreover, based on the electrophoretic mobility, we can obtain detailed information about the conformation changes that determine both folding topology and molecularity in G4 structure. Therefore, we used this method to study RNA G-quadruplex folding patterns.

Due to the fact that NMM is a highly selective quadruplex-specific ligand with a preference to bind to the parallel G4s [35], we applied NMM staining to confirm RNA G-quadruplex formation. Additionally, as we used ThT probe in our fluorescence investigations, we decided to employ ThT to stain the gel and determine the RNA G4s. In PAGE experiments the following RNA oligomers were used: PQS oligomers including two control probes, i.e., 17KW and its variant with mutation in G-rich region named 18KW, the hammerhead ribozyme named HHO as a negative control and three oligo(dT) of various lengths as the mobility markers (named dT and marked on the gel as T35, T45 and T55). PQS oligomers assigned to the third group (2KW, 4KW, 8KW, 10KW and 12KW) were excluded from the PAGE analysis, because no bands on the gels were observed for these variants after ligand staining. The only exception was 5KW oligomer which is suspected to fold into less stable G4 structure. The resultant gels are presented in Figure 5.

The gels were first visualized under the UV light (Figure 5a). As expected, we observed evident bands in the lines of all oligomers, which confirm that the samples were properly prepared and loaded into the wells (Figure 5a). The analysis of the gel stained with NMM dye revealed the bands for 1KW, 3KW, 5KW, 6KW, 7KW, 11KW variants and control 17KW, whereas there were no bands in the line of 9KW, 18KW, HHO and dT. This phenomenon indicates that tested PQS motifs from influenza A virus (except 9KW) adopt RNA G-quadruplex folds. Moreover, these oligomers demonstrate different electrophoretic mobility, which indicates differences in their folding molecularity. The number and position of bands in the lines of 1KW, 5KW, 7KW, 11KW and 17KW showed that these structures are probably composed of more than one strand, suggesting bimolecular or tetramolecular G-quadruplexes. Furthermore, as presented in Figure 5b, we noted that for the 1KW, 5KW, 7KW, 11KW and 17KW variants appeared few bands corresponding probably to higher-order G4 structures or suggesting that the equilibrium between different conformations exists. In contrast, only one band in the line of 3KW and 6KW was observed (Figure 5b). This result shows that these PQS motifs can form bimolecular G-quadruplex.

According to the gel stained with ThT dye presented in Figure 5c, we noted that the same bands as in the case of NMM staining were observed, however, there are also many additional bands visible. The bottom panel of the gel demonstrated similar bands to those visualized under the UV light (Figure 5a vs. Figure 5c). In this regard, we concluded that ThT ligand is less specific towards G-quadruplexes in comparison to NMM dye. These findings stay in accordance with research showing that ThT can also bind to other non-canonical structures, such as duplexes or triplexes as well as to certain trinucleotide repeats forming mismatches [34,36]. Nonetheless, ThT staining assay served as additional confirmation that most of the selected PQS oligomers fold into RNA G4s.

In summary, all the results obtained by using different spectroscopic methods and electrophoretic techniques, allowed us to discover that three PQS motifs from the influenza A/California/07/2009(H1N1) genome adopt RNA G-quadruplex structures. Nonetheless, these G4s can differ in their stability and also folding topology depending on the experimental conditions.

## 3. Discussion

Influenza virus continuously remains a serious threat for public health. The number of annual deaths worldwide caused by infection with this pathogen reaches above 600,000 cases. Due to the high viral genetic variability, new vaccines need to be developed every year and their effectiveness rarely exceeds 60% [37]. Therefore, development of new therapeutic strategies and the discovery of potential drug targets in vRNA secondary structure are important topics.

Based on the literature we know that the secondary structure of vRNA is highly conserved among viral strains and biologically important for virus life cycle [7]. The viral replication, RNA packaging or its recognition by the host immune system are controlled by the RNA structure. It has been discovered that several short-range vRNA secondary structural motifs are likely to play an important role in virus replication [38]. Moreover, the long-range interactions formed by the partially complementary 5′ and 3′ ends of vRNA are crucial for the regulation of viral transcription and replication [39].

In virus particles and in cells, the vRNA segments are encapsidated as individual RNP complexes. The molecular model of interaction between the vRNA and NP revealed that RNA can loop out from this complex and interact with other vRNA segments or host and virus factors [8,40,41]. Furthermore, in 2018 Williams et al. showed that some regions of vRNA interact more weakly with NP, allowing them to adopt secondary or tertiary RNA structures [41]. Interestingly, Lee and co-workers employed high-throughput sequencing of RNA isolated by the crosslinking immunoprecipitation (HITS-CLIP) methodology and revealed a distinct pattern of NP-association with vRNA [9]. Certain regions of vRNA exhibit strong NP association (peaks), whereas other regions are depleted of NP or can associate and disassociate from NP (non-peaks). Interestingly, they observed a preference in nucleotide composition for NP binding sites, which are relatively guanine-rich and uracil-poor compared to the overall genome-wide nucleotide content. It suggests that vRNA nucleotide content can be an important factor for the degree of NP binding. Moreover, it can be concluded that some regions of influenza vRNA segments not bound by NP would form functional secondary structure and be involved in RNA–RNA interactions [9].

Furthermore, taking into account described by Lee et al. NP-vRNA association profiles (including nucleotide content of NP peaks) in the A/California/07/2009 strain, we can suggest that PQS motifs may occur in G-rich regions that are NP binding positions on the vRNP. However, NP association does not exclude RNA–RNA interactions between viral RNA segments, as reported by Le Sage et al. [42]. Due to the fact that vRNA secondary structure is partially unwound by binding NP, we can postulate that this phenomenon could have an influence on the G4 structure formation in virion or infected cells. On the other hand, it is possible that G4-specific ligand binding to vRNA G-quadruplex can partially inhibit NP association with G-rich regions or induce conformational changes in the RNA structure imposed by NP.

Interestingly, previous research revealed that potential G4 sequences can be found in Zika virus and Zaire ebolavirus (EBOV) genomes and their potential biological functions were studied [17,18]. Burrows and co-workers determined that the formation of G4s by the conserved Zika virus sequence causes polymerase stalling [18]. On the other hand, Wang et al. reported that a highly conserved G-rich sequence present in the EBOV L gene can fold into RNA G4 structure, stabilization of which by the specific ligand leads to repression of this gene [17]. A similar approach has been applied against hepatitis C virus (HCV) [43]. The authors identified a highly conserved G-rich sequence located within the negative RNA strand and revealed that stabilization of formed G4s by Phen-DC3 ligand causes inhibition of HCV viral replication [43].

In this present study, we used bioinformatical and biophysical methods to search the influenza A/California/07/2009(H1N1) genome for PQS motifs and confirm the G4 structure formation. There are many PQS searching bioinformatics tools reviewed by Lombardi et al. [44]. Here, we used two algorithms: G4RNA screener [45] and QGRS mapper [46] to identify and select PQS sites within the influenza A/California/07/2009(H1N1) genome. We found out that 12 PQS motifs are present in the few segments of the IAV genome. It should be noted that one sequence identified in our study, 12KW, was previously found also by Brázda et al. [20], while other potential quadruplex-forming sequences selected by our research groups varied. A possible explanation for this observation is that we identified PQS from influenza A virus vRNA segments using QGRS mapper and G4RNA screener, while Brázda et al. utilized in this purpose G4hunter Web tool. Moreover, in our study we focused on the influenza A/California/07/2009(H1N1) genome analysis and Brázda et al. analyzed G4-EA-H1N1 genomes. The discrepancy can be also related to the fact that different algorithms were used and the settings or parameters selected for searching varied between our and Mergny’s groups. It had an impact on the number of sequences retrieved from the databases as well. In addition, the algorithms used do not consider possible substitutions of G residue by A residue in quadruplex tetrads that have been reported by Kocman and Plavec [47], stable G4 structures with long loops described by Guédin et al. [48], or G-quadruplex with unique features. Therefore, after bioinformatics analysis, the G4 formation by PQS candidates must be confirmed by experimental validation.

Here, we also analyzed the similarity of PQS motifs across the IAV genomic sequences retrieved from the IRD database. Additionally, the PQS conservation analysis based on LOGO sequences showed that G residues within G-tracts, which could be involved in G4 folding, were conserved. Recently, it has been reported that G4-prone sequences from the genomes of RNA viruses show high level of conservation among different genotypes, suggesting their crucial role in the virus life cycle [17,18,43]. For instance, the G-quadruplex formation can inhibit RNA synthesis by the RdRp in HCV negative RNA strand [43].

More recently, some papers described bioinformatics analyses of viral genomes (including human coronaviruses, [49,50] and influenza virus, [20]) to determine the presence of G-rich regions. In 2020, Bartas et al. conducted a systematic and comprehensive analyses to study the occurrence of PQS sites and inverted repeats (IRs) loci in genomes of all known *Nidovirales* [50]. The authors discovered that the G-quadruplex-forming sequences are very rare and unequally distributed in *Nidovirales* genomes, in contrast to IRs. However, location of PQS sites before 3′ UTR, inside 5′ UTR and before mRNA suggests their important regulatory role [50]. Another study concerning G-quadruplex in H1N1 influenza genomes was performed by Brázda et al. [20]. The comparison of 77 H1N1 influenza genomes provided information on G-quadruplex occurrence, localization and variance. The results showed that influenza genomes contain several highly conserved PQS, which can be potential therapeutic targets [20]. Interestingly, in silico analysis of PQS sites in the genome of all the known human-infecting viruses was reported by Lavezzo and co-workers [49]. Comprehensive analysis revealed that PQS sites presence and location are characteristic for each virus class and family. Furthermore, the researchers noticed that PQS motifs are mainly present in the ssRNA viral genomes that are more suitable to G4 structures’ formation as they do not require unfolding from a complementary sequence [49]. Moreover, data indicate that most of the virus classes could regulate their genome functions by G4-mediated mechanism [49].

Our studies focused on the identification of PQS motifs and biophysical characterization of RNA G4s formation. We combined several experimental methods that are commonly used for studying the G-quadruplexes [17,18,26,27,33]. A similar investigation was performed for human coronaviruses [51]. All seven analyzed coronavirus genomes have been demonstrated to contain G4 sequences, and SARS-CoV and SARS-CoV-2 harbor the most conserved sequences [51]. This observation suggests that G-quadruplexes can be important elements in the genomes of human coronaviruses. Formation of G4 structures was experimentally confirmed using fluorescence ThT assay and CD spectroscopy [51], which were also used in our studies.

The growing interest in the G-quadruplex structures led to the discovery of many advantageous properties of these unique forms. Development of novel biosensors based on the G4s with untypical folding features or improved binding affinity reviewed by Xi et al. [14] is one of such examples. The other study presented a fluorescence method for detection of the IAV DNA sequence based on the G4-NMM complex and assistance-DNA [52]. The quadruplex-based functional DNA was designed and used for the homogeneous detection of influenza A DNA sequence [52]. Nevertheless, the current coronavirus disease 2019 (COVID-19) outbreak encouraged many scientists to search for new antiviral drugs and develop novel strategies against viral infections. One example can be the study concerning RNA G-quadruplex in SARS-CoV-2 as a potential therapeutic target [53]. The researchers revealed that PQS in SARS-CoV-2 can form stable RNA G4 structures in live cells. Furthermore, they presented that the SARS-CoV-2 N protein levels both in vitro and in vivo were reduced by a specific targeting compound—PDP (pyridostatin derivative) [53].

In conclusion, a combination of bioinformatics tools, spectroscopic techniques and gel electrophoresis method allowed us to achieve a global view of RNA G-quadruplex formation within the sequences of the IAV genome. Results obtained from our experiments are summarized in Table 4. Based on these data we were able to confirm three PQS motifs (1KW, 7KW and 11KW) to fold into G4s.

To the best of our knowledge, the current investigations provide the first evidence that conserved PQS motifs are present in the influenza A/California/07/2009(H1N1) genome and have a high propensity to fold into RNA G-quadruplex structures. Comparing our investigations with an already published study on the genotype 4 reassortant Eurasian avian-like H1N1 virus (*G4*-EA-H1N1) and the PQS occurrence in *G4*-EA-H1N1 genomes [20], we focused on different approaches, but came to the same conclusion that the IAV genome contains specific G-rich sequences, which can adopt a G-quadruplex fold. Further studies are necessary to confirm the G4s formation in live cells and to obtain a better understanding of the potential role in virus biology, especially taking into the account the fact that the PQS motifs were found within PB1, PB2 and PA segments. Nevertheless, the current study provides information concerning RNA G4 structure that could be explored for use as a novel therapeutic target in the future.

## 4. Materials and Methods

### 4.1. Identification of PQS Motifs in the IAV Genome

The influenza A/California/07/2009(H1N1) genome was obtained from the NCBI database (RefSeq assembly accession: GCF_001343785.1). To identify the PQS motifs, the genome was analyzed using two different tools for G-quadruplex prediction, i.e., G4RNA screener web interface (accessed on 15 December 2020) (http://scottgroup.med.usherbrooke.ca/G4RNA_screener, v.0.3, Scott Group Bioinformatics, Sherbrooke, QC, Canada) and the Quadruplex forming G-Rich Sequences (QGRS) Mapper (accessed on 15 December 2020) (http://bioinformatics.ramapo.edu/QGRS/analyze.php, Ramapo College of New Jersey, Mahwah, NJ, USA).

### 4.2. Sequence Alignment and Frequency of Nucleotide Residues Analysis

The nucleotide sequences for each segment of the IAV were downloaded from the Influenza Research Database (IRD) (accessed on 15 December 2020) (www.fludb.org). Two types of comparative analysis of sequence data were conducted with the help of MAFFT software (version 7, Research Institute for Microbial Diseases, Suita, Osaka, Japan) and the Bioedit 7.1 program (version 7.1, Thomas A. Hall, Vista, CA, USA). The multiple sequence alignments using MAFFT were performed for all IAV genomic sequences deposited in the IRD database and separately for all those of IAV H1N1 subtypes. Next, the aligned sequences were loaded into the Bioedit 7.1 program for the frequency of nucleotide residues analysis. The data obtained were used to examine the PQS motifs similarity across both the IAV subtypes and particularly the IAV H1N1 subtype. Then PQS motifs were searched in aligned segment sequences, LOGO representations were constructed using WebLogo application (accessed on 15 December 2020) (https://weblogo.berkeley.edu/logo.cgi, version 2.8.2, Department of Plant and Microbial Biology, University of Berkley, Berkley, CA, USA) and the conservation levels were determined.

### 4.3. Synthesis and Purification of Oligonucleotides

RNA oligonucleotides were synthesized on a MerMade12 (BioAutomation, Irving, TX, USA) synthesizer using standard phosphoramidite chemistry on solid support [54]. All oligonucleotides were purified by polyacrylamide gel electrophoresis. The details of deprotection and purification of oligonucleotides have been described previously [55].

### 4.4. NMR Spectroscopy

^1^H NMR spectra were acquired on a Bruker AVANCE III 700 MHz spectrometer equipped with a QCI CryoProbe (Bruker, Billerica, MA, USA). The 3 mm thin wall tubes were used with the final sample volume of 200 μL H_2_O/D_2_O (9:1, v/v). RNA oligonucleotides were dissolved in a 10 mM potassium phosphate buffer (pH 6.8) containing 50 mM KCl and 0.1 mM EDTA (ethylenediaminetetraacetic acid). The experiments were conducted at 25 °C using RNA oligomers with or without the addition of N-methyl mesoporphyrin IX (NMM, Frontier Scientific, Logan, UT, USA) in a 1:1 molar ratio. The samples were annealed by heating at 90 °C for 5 min and then slowly cooled to room temperature. Water suppression was achieved using excitation sculpting. The spectra were processed and prepared with TopSpin 3.0 Bruker Software (version 3.0, Bruker, Billerica, MA, USA).

### 4.5. Ultraviolet (UV) Melting Measurements

RNA oligonucleotides were dissolved in the same buffer as used for NMR studies. Oligonucleotide single strand concentrations were calculated based on both, absorbance measured above 80 °C and extinction coefficients, which were approximated by a nearest-neighbor model using the Integrated DNA Technologies calculator on the website https://www.idtdna.com/calc/analyzer (Integrated DNA Technologies, Inc., Coralville, IA, USA). Samples were denatured for 5 min at 90 °C and then slowly cooled to room temperature. Measurements were performed in triplicate using quartz cuvettes of 0.1 cm path length with a sample volume of 30 µL. Absorbance versus temperature curves were obtained by the UV melting method at 260 nm and 295 nm wavelengths in the temperature range 5–90 °C with a heating rate of 0.2 °C/min on a Jasco V-650 spectrophotometer (Jasco Deutschland GmbH, Pfungstadt, Germany) equipped with a thermoprogrammer. Thermal denaturation curves were analyzed and the melting temperatures (T_m_) of model PQS oligomers were determined. The T_m_ values were obtained as an arithmetic average from three repeats.

### 4.6. Circular Dichroism Spectroscopy

CD spectra were recorded on a Jasco J-815 spectropolarimeter (Jasco Deutschland GmbH, Pfungstadt, Germany) using 1.5 mL quartz cuvettes with a 5 mm path length and the sample volume of 1300 µL. RNA oligonucleotides were dissolved in the same buffer as used for NMR studies, to achieve a sample concentration of 14 µM. All samples were denatured for 5 min at 90 °C and then slowly cooled to room temperature overnight before data collection. Measurements were collected at 10 °C in the 220–340 nm wavelength range with a 1 nm data interval. CD curves were established as an average of three CD measurements. The buffer spectrum was subtracted from the sample spectra. CD spectra were expressed as the difference in the molar absorption (Δε, in units of cm^2^ mmol^−1^) of the right-handed and left-handed circularly polarized light and normalized for plotting and comparative purposes using Origin Pro 9.8 software (version Origin Pro 2021 (9.8), Northampton, MA, USA).

### 4.7. Fluorescence Intensity Measurements

RNA oligonucleotides at concentration of 1 µM were dissolved in the same buffer as used for NMR studies, to obtain a final volume of 150 µL. Samples were denatured for 5 min at 90 °C and then cooled down to room temperature overnight. After folding, the sample solutions were mixed with thioflavin T (ThT, Sigma Aldrich, Saint Louis, MO, USA) at concentration of 0.5 µM. Next, the samples were centrifuged and loaded into a 96-well plate (Greiner Bio-one, Kremsmünster, Austria) in three repeats. The fluorescence measurements were carried out with CLARIOstar Plus multimode plate reader (BMGLabtech, Ortenberg, Germany) at room temperature. ThT fluorescence emission spectra were acquired over the range of 448 to 718 nm. The excitation wavelength was 425 nm. The fluorescence intensity of ThT was measured at 482 nm with excitation at 425 nm. The data analysis was performed using the Origin Pro 9.8 software (version Origin Pro 2021 (9.8), Northampton, MA, USA).

### 4.8. Native Polyacrylamide Electrophoresis

Native PAGE was conducted to monitor the mobility of RNA oligomers and confirm the RNA G-quadruplex formation. RNA oligonucleotides at concentration of 0.4 µg/mL (absorbance of 0.1 at 260 nm) were dissolved in the same buffer as used for NMR studies, to obtain final volume of 4 µL. Samples were denatured for 5 min at 90 °C and then slowly cooled to room temperature. After folding, 2 μL of 35% glycerol solution was added. Samples were centrifuged and loaded into a 20% native polyacrylamide gel prepared in 1X TBE (Tris-Borate-EDTA) buffer, pH 7.0. The electrophoresis was performed at 150 V for 3 h in a cold room using XCell SureLock Mini-Cell Electrophoresis System (ThermoFisher Scientific, Waltham, MA, USA). For analysis the gel was stained in a solution of 0.1 mg ml^−1^ NMM in 1X TBE for 10 min or in solution of 0.5 µM ThT in 1X TBE for 15 min. After staining, the gels were destained for 10 min in 1X TBE. The resultant gels were visualized with the help of Fujifilm Phosphorimager (Fujifilm Holdings Corporation, Tokio, Japan) and analyzed using MultiGauge Fujifilm software (version 3.0, Fujifilm Holdings Corporation, Tokio, Japan).

## Figures and Tables

**Figure 1 ijms-22-06031-f001:**
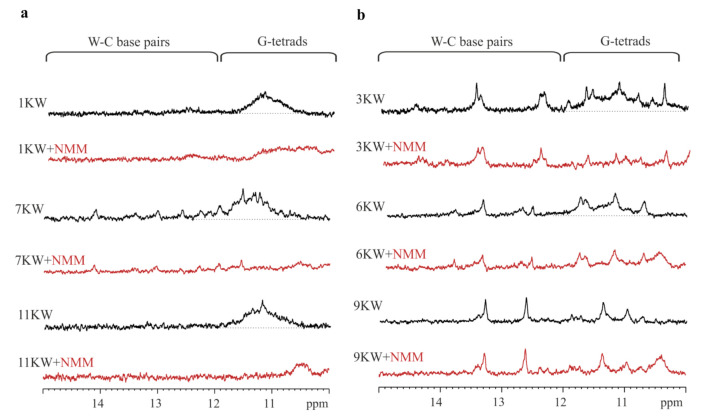
The imino region of the nuclear magnetic resonance (^1^H NMR) spectra of PQS oligomers. (**a**) ^1^H NMR spectra for 1KW, 7KW and 11KW variants (first group), (**b**) ^1^H NMR spectra for 3KW, 6KW and 9KW variants (second group).

**Figure 2 ijms-22-06031-f002:**
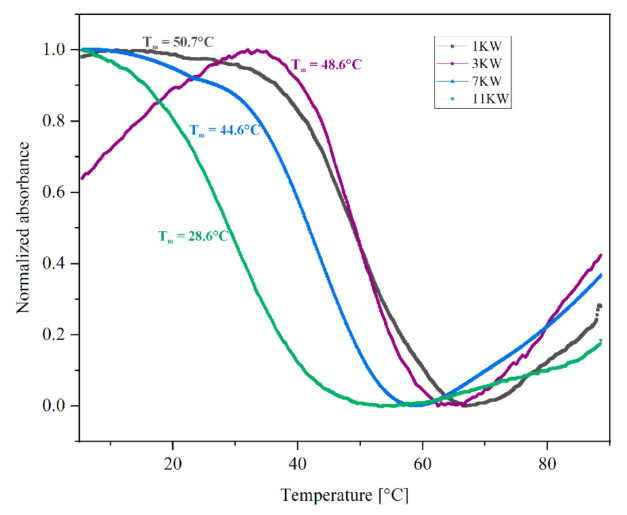
Representative ultraviolet (UV) melting curves of model PQS oligomers with T_m_ values obtained by monitoring the melting profile at 295 nm.

**Figure 3 ijms-22-06031-f003:**
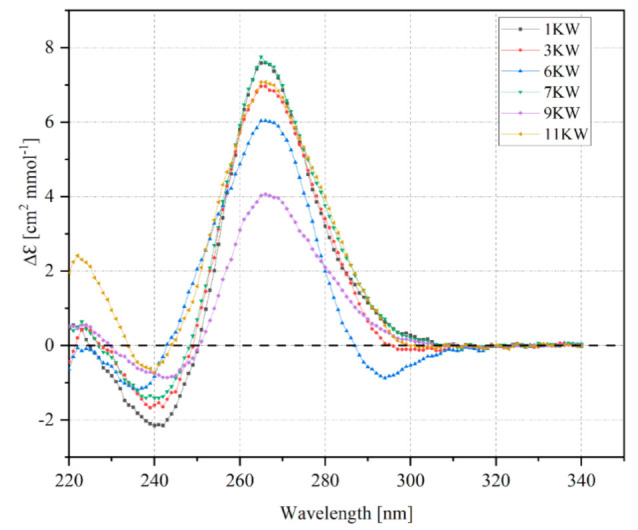
Circular dichroism (CD) spectra of model PQS oligomers recorded at 10 °C.

**Figure 4 ijms-22-06031-f004:**
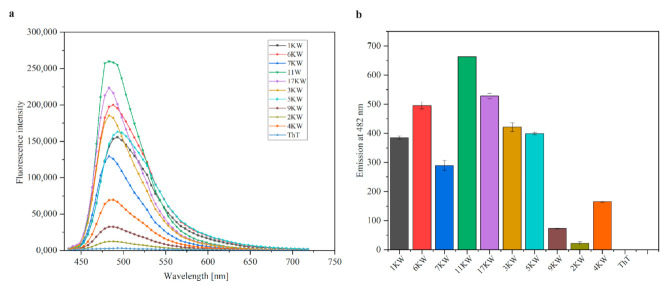
Thioflavin T fluorescence assays for PQS oligomers. (**a**) Fluorescence emission spectra of ThT in the presence of model PQS oligomers, fluorescence of ThT alone is also presented, (**b**) fluorescence emission of ThT at 482 nm for model PQS oligomers compared to ThT alone. The dotted line is a threshold for ThT enhancement supporting G4 formation.

**Figure 5 ijms-22-06031-f005:**
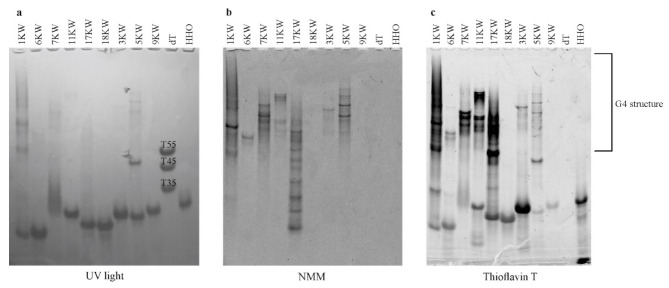
Visualization of model PQS oligomers migration on a native gel. (**a**) The gel under the UV light, (**b**) The gel after staining with G4 specific ligand, N-methyl mesoporphyrin IX (NMM), (**c**) The gel after staining with G4 specific ligand, thioflavin T. On the gels single-stranded dT is a mobility marker and hammerhead ribozyme (HHO) is a negative control.

**Table 1 ijms-22-06031-t001:** Characteristics of potential quadruplex-forming sequences (PQS) motifs from vRNA of the influenza A/California/07/2009(H1N1).

Segment (Protein)	Segment Length (nt)	PQS Name	PQS Location within Segment	Sequence (5′-3′)
1 (PB2)	2341	1KW	436–460	CUGGUGGGGCAGCAGCAAAGGGGAG
		6KW	1018–1040	GG UGCAU GGGG UUCAGUCGCU GG
2 (PB1)	2341	2KW	360–381	UUGGCUGGACCAUGGGCUGGCA
		7KW	2097–2128	GG UAGU GG UCCAUCAAUC G GG UUGAGCU GGGG
		8KW	2224–2259	GGCUGUAUGGAGGAUCUCCAGUAUAAGGGAAUGUGG
3 (PA)	2233	9KW	495–526	UUGGAGGUUCCAUUGGUUCUCACAUAUAGGAA
		10KW	1381–1412	GGCAAAGAGGCCCAUCAGGCAAUCUGAGGGG
		3KW	1532–1564	AAGGCUGGGGAAGUUCGGUGGGAGACUUUGGUC
4 (HA)	1779	11KW	807–834	GG AUGUAUAUUCUGAAAU GGG A GG CU GG
		4KW	1133–1157	UUGGUCAGCACUAGUAGGUGGAUGG
7 (M1) (M2)	1027	12KW	935–959	UCGGCUUUGAGGGGGCCUGACGGGA
8 (NS1) (NS2)	890	5KW	751–778	UCGGCGGAGCCGAUCAAGGAAUGGGGCA
G—G-rich sites which can be involved in RNA G-quadruplex formation.				

**Table 2 ijms-22-06031-t002:** Summary of the PQS motifs similarity within the influenza virus type A and H1N1 subtype strains.

PQS Name	Influenza Type A		H1N1 Subtype	
	Number of Sequences	Similarity [%]	Number of Sequences	Similarity [%]
1KW	20,593	86.1	8793	93.0
2KW	20,430	96.0	8597	96.4
3KW	20,069	93.3	8743	93.0
4KW	20,534	71.0	9081	86.0
5KW	12,902	95.1	5569	96.7
6KW	20,593	86.8	8793	92.4
7KW	20,430	96.1	8597	93.4
8KW	20,430	97.5	8597	97.8
9KW	20,069	86.9	8743	97.4
10KW	20,069	80.0	8743	92.3
11KW	20,534	84.1	9081	95.0
12KW	11,392	95.4	4767	99.6
Number of sequences deposited in the Influenza Research Database (December 2020).				

**Table 3 ijms-22-06031-t003:** PQS motifs presented as LOGO sequences, their conservation and variability levels.

PQS Name	LOGO Sequence	PQS Length (nt)	Conserved Nucleotides	Conservation (%)	Variable Nucleotides	Variability (%)
1KW	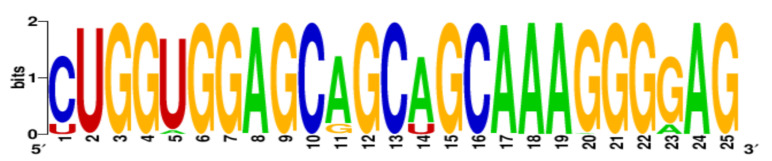	25	20	80.0	5	20.0
2KW	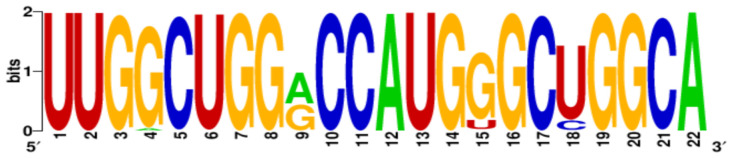	22	18	81.8	4	18.2
3KW	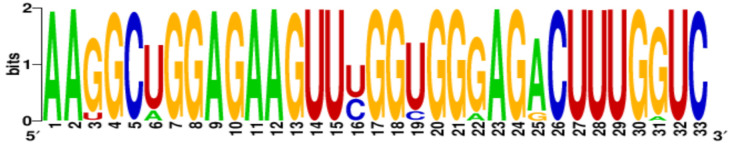	33	26	78.8	7	21.2
4KW	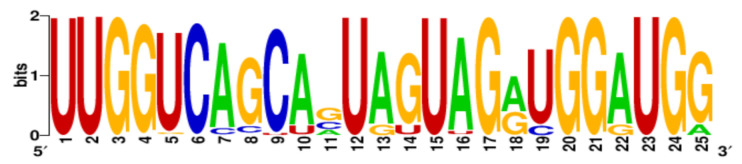	25	11	44.0	14	56.0
5KW	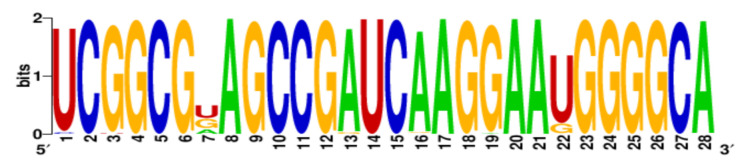	28	21	75.0	7	25.0
6KW	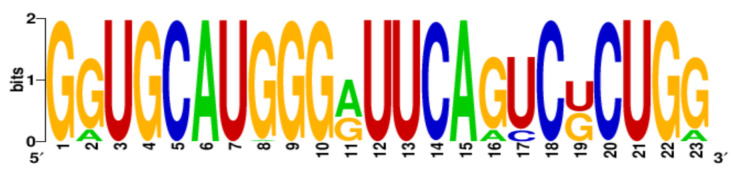	23	16	69.6	7	30.4
7KW	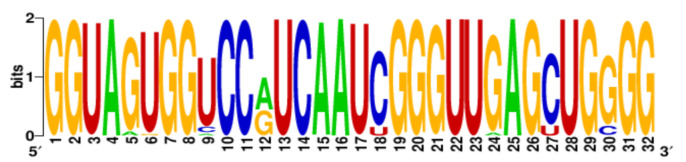	32	24	75.0	8	25.0
8KW	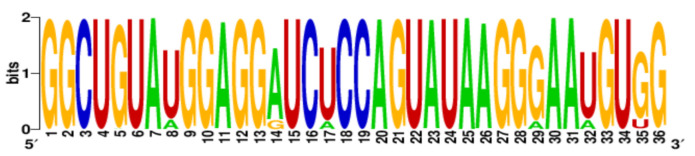	36	30	83.4	6	16.6
9KW	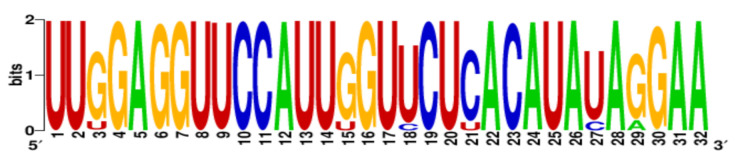	32	26	81.3	6	18.7
10KW	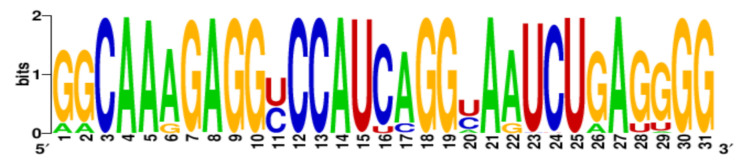	31	20	64.5	11	35.5
11KW	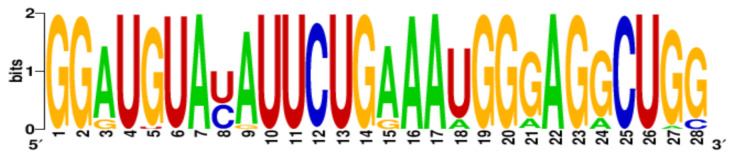	28	18	64.3	10	35.7
12KW	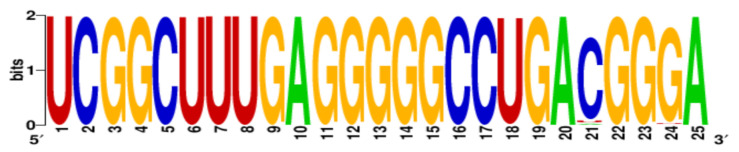	25	23	92.0	2	8.0

**Table 4 ijms-22-06031-t004:** Summary of obtained results with different methods for selected PQS oligomers.

PQS Name	^1^H NMR Spectra	UV Melting Profile at 295nm	CD Spectra	Thioflavin T Fluorescence Enhancement	Native Polyacrylamide Gel Electrophoresis	G4 Structure Folding
1KW	✓	✓	✓	✓	✓	yes
7KW	✓	✓	✓	✓	✓	yes
11KW	✓	✓	✓	✓	✓	yes
3KW	✓ -	✓ -	✓	✓	✓	probably
6KW	✓ -	✓ -	✓ -	✓	✓	probably
9KW	✓ -	✓ -	✓	✓ -	✓ -	no

## Data Availability

The data used to support the findings of this study are available upon request to the authors.

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
