# Peer review of "Identification and Structural Aspects of G-Quadruplex-Forming Sequences from the Influenza A Virus Genome"

_ijms, 2021, doi:10.3390/ijms22116031_

Round 1

Reviewer 1 Report

In the manuscript, the authors employed several approaches (1H-NMR, CD, UV melting, fluorescence spectroscopy and gel) to identify several G-quadruplex forming sequences within the influenza A genome. The manuscript can be accepted if the following minor comments have been addressed:

  1. The authors could try to use CD melting experiment to obtain the thermodynamic parameter and Tm values of the RNA G-quadruplex.
  2. Stabilizing the G-quadruplex structure in virus by small ligand is a potential method to inhibit the translation. The authors should compare the stability of G4-ligand and free G4.

Author Response

Response to Reviewer 1 Comments

In the manuscript, the authors employed several approaches (1H-NMR, CD, UV melting, fluorescence spectroscopy and gel) to identify several G-quadruplex forming sequences within the influenza A genome. The manuscript can be accepted if the following minor comments have been addressed:

Comment 1. The authors could try to use CD melting experiment to obtain the thermodynamic parameter and Tm values of the RNA G-quadruplex.

As described in our manuscript, thermodynamic studies of the RNA G-quadruplexes were conducted using the UV/VIS melting method. This technique is fast, simple and commonly used as the experimental tool to determine the G4s thermodynamic parameters in our laboratory as well as by other researchers, e.g. Jaubert et al. investigated G4s in the HCV negative RNA strand (doi: 10.1038/s41598-018-26582-3), Burrows and co-workers studied G-quadruplex formation from Zika virus genomic RNA (doi: 10.1021/acsinfecdis.6b00109), Ji et al. confirmed G4s in the RNA genome of SARS-CoV-2 using UV melting assay (doi: 10.1093/bib/bbaa114). However, the CD spectroscopy is also a valuable biophysical technique for verifying G-quadruplex formation. It is an alternative to UV/VIS melting method. There are some reports concerning G4s formation by CD melting experiment (e.g. Majee et al., doi: 10.1038/s41598-020-58406-8; Zhao et al., doi: 10.1002/anie.202011419).

We applied CD measurements for studying the folding topology of G4 structures formed by PQS candidates. In our structural analysis the CD spectroscopy constitutes a complementary spectroscopy method. 

More specific, we first performed the UV melting measurements for nine different concentrations of each PQS oligomer in the concentration range 10-4–10-6 M to determine thermodynamic parameters (i.e. enthalpy, entropy and Gibbs free energy). However, we did not observe the melting curve at 295 nm wavelength for each PQS oligomer concentrations, thus we were not able to determine reliable thermodynamic data. It can be related to the fact that in case of more complex structures (such as G-quadruplex) the non-cooperative transition to single-stranded form during UV melting experiment is possible. As a result, we decided to focus on the melting profiles of PQS variants and changes in the melting temperature (Tm parameter). The Tm values were obtained as an arithmetic average from three repeats.

Comment 2. Stabilizing the G-quadruplex structure in virus by small ligand is a potential method to inhibit the translation. The authors should compare the stability of G4-ligand and free G4.

Thank You for this suggestion. It would have been interesting to concern this aspect. However, in the case of our study, the main goal was to examine the genome of the IAV for unique G-rich sequences and confirm the RNA G4 structure formation. Thus, in the first step we identified PQS motifs within IAV genome and then we investigated their ability to form G-quadruplexes using different biophysical methods. We examined thermal stability of G4s (based on the melting temperature) and structural features of PQS using NMR, CD and fluorescence spectroscopies as well as the PAGE experiments. Though, we consider that the stability of RNA G4s can differ upon ligand binding (e.g. small molecules), we did not examine the influence of specific G4 ligand on G-quadruplex stability (melting temperature). We tested two ligands (i.e. NMM and thioflavin T, ThT) selectively binding to G-quadruplexes in: NMR experiments, fluorescence assay and native gel electrophoresis to confirm G4s formation. Based on our data obtained from these experiments, we can postulate that the ligand binding to G-quadruplexes can influence the stability of formed G4s. The fluorescence enhancement resulting from the specific binding of ThT to G4s or evident bands observed on the gels after staining with NMM or ThT can suggest that the G4-ligand association caused the stabilization of these structures.

At present, we are working on biological studies of the PQS ability to form G4s. We will investigate the potential ligand-induced stabilization of G-quadruplexes in vitro and in infected cells. For this purpose, selected ligand - TMPyP4, a porphyrin known to interact with G-quadruplex structure, will be used.

Reviewer 2 Report

Previously, another group has reported G-quadruplex-forming sequences in the influenza A virus sequence. However, the authors identified new sequences that can form G-quadruplex by different approaches and evaluated G-quadruplex formation by multifaceted evaluation. The information in this manuscript will help readers to analyze the role of G-quadruplex for viral propagation. Overall, this manuscript would be acceptable, but I have some suggestions to improve this manuscript. 1. The authors focus on the genome sequence of flu. In virion and cells, viral proteins interact vRNA to form vRNP. The Secondary structure of vRNA is partially unwound by binding NP. The authors do not mention this point in the introduction and discussion session. 2. Related to 1, NP binding positions on vRNP were analyzed by reported CLIP analysis. The information of NP binding at the identified sequences in this manuscript is helpful for a discussion about G-quadruplex formation on vRNAin virion and infected cells. 3. Figure 1. What are the criteria for grouping the NMR data? NMR spectra of 6KW looks in equilibrium between two structures like 3KW and 9KW. 4. 12KW was also isolated in the previous study. This point should be mentioned in the discussion session, and the authors should discuss why the different sequences were identified. 

Author Response

Response to Reviewer 1 Comments

Previously, another group has reported G-quadruplex-forming sequences in the influenza A virus sequence. However, the authors identified new sequences that can form G-quadruplex by different approaches and evaluated G-quadruplex formation by multifaceted evaluation. The information in this manuscript will help readers to analyze the role of G-quadruplex for viral propagation. Overall, this manuscript would be acceptable, but I have some suggestions to improve this manuscript.

Comment 1. The authors focus on the genome sequence of flu. In virion and cells, viral proteins interact vRNA to form vRNP. The Secondary structure of vRNA is partially unwound by binding NP. The authors do not mention this point in the introduction and discussion session.

It needs to be mentioned that the vRNA secondary structure determined under laboratory conditions may differ from that naturally occurring in a virion and an infected host cell. It is because vRNA forms complex called viral ribonucleoprotein (vRNP) that contains multiple copies of the viral nucleoprotein (NP) and viral polymerase proteins. Due to the fact that vRNA function is closely related with its structure, studies concerning the RNP structure and its assembly are important for better understanding of influenza virus structure-function relationships.

To make this point more informative and clear we added to the manuscript the following fragments:

Introduction section [page 2, paragraph 2]:

″Overall, vRNP structure is related to its functions. It was reported that the viral replication, RNA packaging, mRNA splicing regulation or recognition by the host immune system are controlled by the RNA structure [7,8]. Within the vRNP complex NP molecules bind vRNA with high affinity and serve as the regulator of the nuclear export and import of vRNP. However, it was previously confirmed that vRNA can escape from complex with NP and potentially fold into the secondary structure in a dynamic manner [9].″

Discussion section [page 13, paragraph 3]:

″In virus particles and in cells the vRNA segments are encapsidated as individual RNP complexes. The molecular model of interaction between the vRNA and NP revealed that RNA can loop out from this complex and interact with other vRNA segments or host and virus factors [8,40,41]. Furthermore, in 2018 Williams et al. showed that some regions of vRNA interact more weakly with NP, allowing them to adopt secondary or tertiary RNA structures [41].″

Comment 2. Related to 1, NP binding positions on vRNP were analyzed by reported CLIP analysis. The information of NP binding at the identified sequences in this manuscript is helpful for a discussion about G-quadruplex formation on vRNA in virion and infected cells.

We added listed below fragments to the manuscript to make a discussion more informative and emphasize the possible effect of NP binding in relation to the G-quadruplex formation.  

Discussion section [pages 13 and 14, paragraphs 3 and 4]:

″Interestingly, Lee and co-workers employed high-throughput sequencing of RNA isolated by crosslinking immunoprecipitation (HITS-CLIP) methodology and revealed a distinct pattern of NP-association with vRNA [9]. Certain regions of vRNA exhibit strong NP association (peaks), wherease other regions are depleted of NP or can associate and disassociate from NP (non-peaks). Interestingly, they observed a preference in nucleotide composition for NP binding sites, which are relatively guanine-rich and uracil-poor compared to the overall genome-wide nucleotide content. It suggests that vRNA nucleotide content can be an important factor for the degree of NP binding. Moreover, it can be concluded that some regions of influenza vRNA segments not bound by NP would form functional secondary structure and be involved in RNA-RNA interactions [9].

Furthermore, taking into account described by Lee et al. NP-vRNA association profiles (including nucleotide content of NP peaks) in the A/California/07/2009 strain, we can suggest that PQS motifs may occur in G-rich regions that are NP binding positions on the vRNP. However, NP association does not exclude RNA-RNA interactions between viral RNA segments, as reported by Le Sage et al. [42]. Due to the fact that vRNA secondary structure is partially unwound by binding NP, we can postulate that this phenomenon could have an influence on the G4 structure formation in virion or infected cells. On the other hand, it is possible that G4 specific ligand binding to vRNA G-quadruplex can partially inhibit NP association with G-rich regions or induce conformational changes in the RNA structure imposed by NP. ″

Comment 3. Figure 1. What are the criteria for grouping the NMR data? NMR spectra of 6KW looks in equilibrium between two structures like 3KW and 9KW.

Model PQS oligomers were divided into three groups depending on the obtained NMR spectra, i.e. the presence of characteristic signals from the imino protons in the regions of 10-12 ppm (Hoogsteen hydrogen bonds) and 12-15 ppm (Watson–Crick hydrogen bonds). We performed also NMR experiments with the addition of N-methyl mesoporphyrin IX (NMM) that is a dye selectively binding to G-quadruplexes. After NMM binding to the G4 structure, the signals between 10-12 ppm are usually broad and shifted, which indicates NMM-G4s association. 

When we analyzed the obtained NMR spectra of PQS oligomers, we focused on the signals from imino protons characteristic for the canonical W-C base pairs (12-15 ppm) and Hoogsteen hydrogen bonds (10-12 ppm). The spectrum of 6KW displays the signals in the range of 10-12 ppm, which indicates the G4s formation, but we also observe the imino proton peaks between 12 and 15 ppm corresponding to W-C hydrogen bonds. If we take into account the intensity of these signals, we can suggest that the G4 structure is dominant, but the hairpin is also formed (peaks observed within the range of 12-14 ppm). Due to the fact that the 1H NMR spectrum of 6KW is more similar to the 1H NMR spectra of both 3KW and 9KW, we decided to move the 6KW variant to group 2, as suggested by the Reviewer.

Figure 1 was replaced by Figure with 1H NMR spectra for (a) first group (1KW, 7KW and 11KW) and (b) second group (3KW, 6KW and 9KW), respectively (Results section).

We also changed all sentences about the grouping the NMR data in the manuscript (Results section) and corrected the text in the appropriate fragments describing 6KW and its structural features. All changes were made using the track changes and highlighted in yellow.

Comment 4. 12KW was also isolated in the previous study. This point should be mentioned in the discussion session, and the authors should discuss why the different sequences were identified. 

We agree with this comment. One PQS candidate (12KW) predicted in our computational analysis was also identified in the previous study reported by Brázda et al (doi: 10.1186/s12864-021-07377-9). Interestingly, other PQS candidates selected by our research groups varied.

To make this point more clear we added to the manuscript the following fragment:

Discussion section [page 14, paragraph 6]:

″It should be noticed, that one sequence identified in our study, 12KW, was previously found also by Brázda et al. [20], while other potential quadruplex-forming sequences selected by our research groups varied. A possible explanation for this observation is that we identified PQS from influenza A virus vRNA segments using QGRS mapper and G4RNA screener, while Brázda et al. utilized in this purpose G4hunter Web tool. Moreover, in our study we focused on the influenza A/California/07/2009(H1N1) genome analysis and Brázda et al. analyzed G4-EA-H1N1 genomes. The discrepancy can be also related to the fact that different algorithms were used and the settings or parameters selected for searching varied between our and Mergny’s groups. It had an impact on the number of sequences retrieved from the databases as well. In addition, used algorithms do not consider possible substitutions of G residue by A residue in quadruplex tetrads that have been reported by Kocman and Plavec [42], stable G4 structures with long loops described by Guédin et al. [43], or G-quadruplex with unique features. ″

Additionally, as described in Discussion section [page 14, paragraph 6]: ″There are many PQS searching bioinformatics tools reviewed by Lombardi et al. (…) In addition, used algorithms do not consider possible substitutions of G residue by A residue in quadruplex tetrads that have been reported by Kocman and Plavec [42], stable G4 structures with long loops described by Guédin et al. [43], or G-quadruplex with unique features.″